# *Listeria monocytogenes*—How This Pathogen Uses Its Virulence Mechanisms to Infect the Hosts

**DOI:** 10.3390/pathogens11121491

**Published:** 2022-12-07

**Authors:** Jacek Osek, Kinga Wieczorek

**Affiliations:** Department of Hygiene of Food of Animal Origin, National Veterinary Research Institute, 24-100 Puławy, Poland

**Keywords:** *L. monocytogenes*, listeriosis, virulence traits, host infection, molecular mechanisms

## Abstract

Listeriosis is a serious food-borne illness, especially in susceptible populations, including children, pregnant women, and elderlies. The disease can occur in two forms: non-invasive febrile gastroenteritis and severe invasive listeriosis with septicemia, meningoencephalitis, perinatal infections, and abortion. Expression of each symptom depends on various bacterial virulence factors, immunological status of the infected person, and the number of ingested bacteria. Internalins, mainly InlA and InlB, invasins (invasin A, LAP), and other surface adhesion proteins (InlP1, InlP4) are responsible for epithelial cell binding, whereas internalin C (InlC) and actin assembly-inducing protein (ActA) are involved in cell-to-cell bacterial spread. *L. monocytogenes* is able to disseminate through the blood and invade diverse host organs. In persons with impaired immunity, the elderly, and pregnant women, the pathogen can also cross the blood–brain and placental barriers, which results in the invasion of the central nervous system and fetus infection, respectively. The aim of this comprehensive review is to summarize the current knowledge on the epidemiology of listeriosis and *L. monocytogenes* virulence mechanisms that are involved in host infection, with a special focus on their molecular and cellular aspects. We believe that all this information is crucial for a better understanding of the pathogenesis of *L. monocytogenes* infection.

## 1. Introduction

This comprehensive review, based on the most important literature available on the Web of Science Core Collection published until October 2022, aims to summarize the current knowledge on *L. monocytogenes* virulence mechanisms that are involved in host infection, with a special focus on their molecular and cellular aspects. Firstly, the epidemiology of listeriosis is briefly described. Then, key findings related to surface molecules expressed by the pathogens during intestinal infection and cell-to-cell spread as well as dissemination in the host following the brain and placenta colonization are presented. Finally, recent data regarding treatment and prevention of listeriosis in humans are summarized. We believe that all this information is crucial for a better understanding of the pathogenesis of *L. monocytogenes* infection and prevention of this serious food-borne disease.

*Listeria monocytogenes* was first identified by Murray, Webb, and Swann in 1924, as a gram-positive bacillus responsible for epidemic cases of mononucleosis, a disease affecting laboratory rabbits and guinea pigs, and named it *Bacterium monocytogenes* [1]. A few years later, similar bacteria were isolated from humans in Denmark by Nyfeldt, who described that this microorganism was the cause of infectious mononucleosis [2]. Then, *L. monocytogenes* was recognized as a cause of sporadic infections in workers contacted with diseased animals [3]. Currently, *L. monocytogenes* is considered as one of the most dangerous bacterial pathogens responsible for food-borne illnesses in humans [3,4].

At present, the genus *Listeria* consists of 26 species, including 18 *Listeria* species recognized and described for the first time in 2009 [5,6,7]. However, only two species, *L. monocytogenes* and *L. ivanovii*, are considered pathogenic for humans, but the infections of people with the latter one are very rare [8,9]. There is also little information on *L. seeligeri* isolation from human sporadic listeriosis cases [10]. Furthermore, *L. innocua* has been identified in a person with fatal listeriosis, which suggests that at least some strains of this *Listeria* species are pathogenic for humans and may be able to cause the disease [11].

*L. monocytogenes* has been often identified in domestic and wild animals, and especially ruminants are mostly affected, often without any clinical symptoms, but excreting the bacteria in their feces [12]. Bacteria occur ubiquitously in nature and have been detected in surface water, soil, plant, silage, sewage, and food production environments. They are able to grow at various temperatures ranging from −0.4 °C to 45 °C, pH between 4.6 and 9.5, and have the ability to persist in salt concentrations up to 20% [13,14]. These features allow *L. monocytogenes* to survive and multiply under extreme environmental conditions often present at food production facilities [15,16]. Thus, it is one of the most important food-borne pathogens responsible for sporadic infections or disease outbreaks, admittedly with rather low case numbers, but with a high mortality rate of 20–30% worldwide [17]. The food-connected way of *L. monocytogenes* infection in humans was unclear until the 1980s, when, during several outbreaks in the USA and Switzerland, various foods including dairy products, meat products, seafood products, and vegetables were indicated as the source of bacteria [18,19]. Infection via direct contact with animals or humans with listeriosis or with contaminated environments is also possible, although it is very rare [20,21,22].

The incidences of listeriosis worldwide are very low compared to other food-borne infections, but it is a disease with a very high hospitalization rate of over 95% and high mortality [23]. The recent EFSA and ECDC report for the year 2020 showed that 1876 laboratory confirmed invasive listeriosis cases in humans were noted in the European Union (EU), with the notification rate of 0.42 cases per 100,000 population [4] (Figure 1). The disease was most commonly reported in the age group over 64 years, covering 72.5% of all cases (1360 persons) [4]. Among all infected persons, 780 required hospitalization and 167 died, mainly patients of the age 64–84 years (58.1%) and over 84 years (20.8%) [4]. The number of listeriosis cases, the incidence of the disease per 100,000 population, and the number of deaths in the EU countries during 2011–2020 are shown in Figure 1. In the USA, the Centers for Disease Control and Prevention estimate that about 1600 people are suffering from invasive *L. monocytogenes* infection each year, with the hospitalization rate of ca. 94% and about 260 deaths [23].

The infection with *L. monocytogenes* is especially dangerous for the elderly, pregnant women, unborn babies, and persons with impaired immune systems, e.g., those with AIDS or cancer, or people after transplantation of different organs [24]. It has been estimated that persons with AIDS have a 500 times higher chance to fall ill with listeriosis compared to immunocompetent people of the normal population [25].

The incubation time of listeriosis varies widely according to the clinical form of the disease, from one day, when the non-invasive variants occur, to even 70 days during the invasive form [21,26]. A longer period is observed for pregnancy-associated cases (range 17–67 days) than for central nervous system-associated infections (range 1–4 days) and for bacteremia (range 1–12 days) [26]. Infection of healthy persons with *L. monocytogenes* often manifests as diarrhea with febrile, usually self-limited or may be also asymptomatic; whereas in some individuals, especially those who are immunosuppressed, it can cause invasive listeriosis [27]. However, most patients recover after 24–72 h without any medical treatment [28,29]. The frequency of non-invasive gastrointestinal listeriosis is difficult to establish, mainly because of its unspecific symptoms [12].

Generalized infections due to *L. monocytogenes* affect about 20–40% of patients with the invasive form of the disease. The clinical symptoms of septicemia, such as fever, myalgia, and general malaise, are similar to other etiologic agents causing bacteremia [30]. The infected persons often appear severely ill with fever, nausea, vomiting, and malaise [30,31]. This form of listeriosis is usually complicated with disseminated intravascular coagulation and multi-organ failure such as myocarditis and valvular endocarditis, hepatitis, and endophthalmitis [7,32]. Eventually, the infection can develop into septic shock and death [20]. The mortality rate has been estimated at 20–30% [27,33].

In adults with the invasive form of listeriosis, the most common clinical signs are related to meningitis, due to the bacterial tropism to the central nervous system [27]. According to surveillance data, around 20–25% of *L. monocytogenes* infection cases resulted in invasion of the bacteria to the central nervous system (CNS) [34,35,36]. Other epidemiological data show that human neurolisteriosis occurs in up to 79% of non-perinatal and 19% of perinatal cases, with mortality rates from 17% to 30% [37,38].

## 2. Pathogenesis of Listeriosis

The schematic *L. monocytogenes* infectious cycle in the human host is shown in Figure 2, and the main virulence factors involved in pathogenesis of infection are presented in Table 1. Listeriosis is mainly a result of ingestion of contaminated food, and it is a typical food-borne disease, where the gastrointestinal tract is the primary site of entry of the microorganism into the host [7,20,39]. However, the disease can sometimes be a result of either local infection of other surfaces (the cornea and conjunctiva of the eye or skin) or the genital tract [21,32]. The infective dose of *L. monocytogenes* for humans is difficult to experimentally assess, but it has been estimated at 10^4^ to 10^7^ cells in susceptible persons (e.g., immunocompromised people), to more than 10^7^ bacteria in healthy individuals [40,41,42]. However, during the listeriosis outbreak in the USA identified in 2015, the prevalence of *L. monocytogenes* was estimated at even less than 100 cells in 1 gram of consumed ice cream [41,43]. It generally seems that *L. monocytogenes* is less pathogenic compared to other food-borne bacterial pathogens, which has been confirmed in a mouce animal model where LD_50_ values for orally and parenterally infected animals were estimated at 10^9^ and 10^6^ cells, respectively [44,45,46].

### 2.1. Passage through the Gastrointestinal Tract

*L. monocytogenes* can colonize the gastrointestinal tract due to its resistance to gastric and biliary acids [47] (Figure 2). A low pH environment, present both in the stomach and duodenum, makes a significant barrier for *L. monocytogenes* [48,49]. It has been shown that persons treated towards reduction of gastric acid secretion (e.g., with proton pump inhibitors) represent a higher risk of invasive listeriosis [50]. However, the bacteria are able to adapt to this low pH environment, utilizing different cellular and molecular mechanisms (Table 1).

**Table 1 pathogens-11-01491-t001:** *L. monocytogenes* main virulence traits involved in pathogenesis of listeriosis.

Virulence Gene	Gene Product/Function	Function	Reference
Survival in the intestine			
*gadD1-gadD3*	Glutamate decarboxylase (GAD) system	Acid tolerance	[51,52]
*arcABC*	Arginine deiminase (ADI) pathway		[53,54]
*bsh*	Bile salt hydrolase	Bile tolerance	[55]
*bilE*	Bile exclusion system		[56]
*sigB*	General stress sigma factor		[57]
Epithelial cell adhesion			
*lmo1413*	*Listeria*-mucin-binding invasin A	Intestinal mucus penetration	[58]
*inlA*	Internalin A (InlA)	Adhesion and entry to enterocytes	[59]
*inlB*	Internalin B (InlB)		[60,61]
*lap*	*Listeria* Adhesion Protein (LAP)		[62,63]
*inlP1; inlP4*	Internalin P1 (InlP1); internalin P4 (InlP4)	Invasion and intracellular proliferation	[64]
Dissemination in the host			
*inlF*	Internalin F (InlF)	Adhesion and invasion of macrophages	[65]
*hly*	Listeriolysin O (LLO)	Escape from *L. monocytogenes*-containing vacuoles	[66]
*ifi30*	Gamma-interferon Inducible Lysosomal Thiol reductase (GILT)		[67]
*pplA*	Peptide pheromone-encoding lipoprotein A (PplA)		[68]
*actA*	Actin assembly-inducing protein (ActA)	Cell-to-cell spread	[69,70]
*inlC*	Internalin C (InlC)		[71]
*vip*	Cell invasion LPXTG protein (Vip)	Brain colonization	[72]
*inlF*	Internalin F (InlF)		[73]
*inlB*	Internalin B (InlB)		[74]
*inlP*	Internalin P (InlP)	Placenta colonization	[75]

Several studies have shown that in vitro adaptation of *L. monocytogenes* to acidic pH of 5.5 for 2 h induces the tolerance to higher acid concentrations, including resistance to lethal acidic shock [76,77,78]. This process is a result of the bacteria increasing cytoplasmic buffer capacity through either the glutamate decarboxylase (GAD) system or the activity of an internal proton pump [51,77]. It has been shown that the GAD system is one of the major mechanisms responsible for the maintenance of the bacterial intracellular homeostasis [51]. The whole GAD mechanism is encoded by five genes, responsible for expression of decarboxylases (*gadD1*, *gadD2*, and *gadD3*) and production of antiporters (*gadT1* and *gadT2*) [52]. All these genes are localized at the listerial stress survival islet (SSI-1) [78]. Interestingly, this islet also encodes another gene, a putative penicillin V acylase, that is required for *L. monocytogenes* bile tolerance [56]. It has been shown that isolates with deleted SSI-1 were significantly affected in terms of growth in foods and biofilm formation, suggesting an important role of this islet in survival in adverse environmental conditions, including the human gastrointestinal tract [79].

Recently, Bai et al. [80] demonstrated that *L. monocytogenes* with biofilm-forming ability showed a significantly decreased adhesion and invasion to Caco-2 and HCT-8 cells in vitro compared to that of their planktonic counterparts. Furthermore, such sessile cells also showed significantly lower transepithelial translocation than the planktonic *L. monocytogenes* variants and, thus, possibly have a reduced virulence potential. The authors have also shown that planktonic strains caused significantly more cell damage than the corresponding biofilm-isolated cells [80]. Additionally, it has been demonstrated that the biofilm-isolated *L. monocytogenes* possessed a temporarily attenuated capacity to translocate across the gut barrier and to disseminate in the blood circulation during the early phase of infection (12–24 h), while both planktonic and biofilm-isolated bacteria were able to disseminate to extra-intestinal tissues similarly at 48 h in a murine model due to reduced expression of regulatory proteins (listeriolysin regulatory protein, PrfA, and the general stress sigma factor σ^B^) and virulence factors (InlA, listeriolysin O) [80].

Another important cell system that protects *L. monocytogenes* from adverse environmental low pH is connected with the arginine deiminase (ADI) pathway, in which two other enzymes, carbamoyltransferase and carbamate kinase, encoded by the *arcABC* operon, are involved [54] (Table 1). During this action, ammonia as a byproduct binds to intracellular protons to produce NH4^+^ and maintain the cytoplasmic pH, thereby protecting the *L. monocytogenes* cell from adverse acidic extracellular environments [53,79,81]. Interestingly, the ADI pathway is not active in the non-pathogenic *Listeria* species, e.g., in *L. innocua* [48]. Both GAD and ADI pH adaptation systems in *L. monocytogenes* act simultaneously and make the bacteria able to survive and adapt to acid stress conditions present in the gastrointestinal tract [47,48].

A further adverse gastrointestinal component to overcome by *L. monocytogenes* during host infection is bile, which is produced in the liver and stored and concentrated in the gallbladder [55]. Bile contains several components such as ions, cholesterol, proteins, bile salts, and pigments [82]. Among them, the most important in terms of *L. monocytogenes* survival in the intestine are bile acids that play a role in the disruption of the bacterial cell wall and membrane proteins and induce DNA damage and oxidative stress in bacterial cells [55]. The molecular mechanisms involved in the protection of *L. monocytogenes* from the adverse action of bile acids and the possible role of bile in the development of listeriosis is not fully understood yet [55]. It is known that bacteria excrete a bile salt hydrolase (Bsh), which is the enzyme that removes some amino acids from the bile salts, decreases their antibacterial activity, and has a positive influence on the bacterial survival in the gastrointestinal tract [55,83]. The expression of the bile salt hydrolase encoding gene (*bsh*) in *L. monocytogenes* depends on the general stress sigma factor σ^B^, which can activate several protective genes during stressful external conditions [84,85]. It has been shown that *L. monocytogenes* that lack the *bsh* gene show decreased resistance to bile in vitro, which results in a lower bacterial number in the feces of orally infected guinea pigs as well as reduced virulence potential and liver colonization of such bacteria after intravenous inoculation of mice [86]. Moreover, the *bsh* gene and bile salt hydrolase activity are present in all human pathogenic *Listeria* strains, which may suggest a relationships between *L. monocytogenes* resistance to bile salts and its ability to colonize the intestinal surface and develop into listeriosis [86].

It has been also shown that *L. monocytogenes* housekeeping sigma factor σ^A^, which generally directs expression of genes required for normal bacterial growth, plays a compensatory role in the absence of σ^B^ under bile exposure [87].

Another bile resistance mechanism identified in *L. monocytogenes* is connected with bile exclusion protein (BilE) regulated under the *prfA* virulence gene, which encodes the expression of listeriolysin regulatory protein PrfA [56]. The BilE protein action against bile also depends on the mentioned σ^B^ regulon, which moderates the activity of PrfA during the host infection [57,88,89].

### 2.2. Epithelial Cell Adhesion

After overcoming host gastrointestinal barriers related to low pH and adverse bile activity, *L. monocytogenes* adheres to respective epithelial cell receptors and enters non-phagocytic epithelial cells [90]. Bacteria possess several different factors that allow them to penetrate the mucus layer of the intestine, composed mainly of mucins secreted by the goblet cells, e.g., the *Listeria*-mucin-binding invasin A (Lmo1413) and internalins (InlB, InlC, InlJ, and InlL) [58,91,92] (Table 1). These virulence factors are covalently attached to the cell wall peptidoglycan via the putative peptidoglycan-bound LPXTG protein motif and facilitate bacterial adhesion or invasion of the host cells [93,94].

*L. monocytogenes* possesses the ability to pass through intestinal, blood–brain, and fetoplacental host physiological barriers, but the gastrointestinal tract is the primary route of infection [95]. The bacteria may use different routes to pass through the intestinal mucosa [96]. One of them is transcytosis through the invasion of goblet cells and enterocytes of the small intestinal villi [43,97]. To reach the surface of the epithelial cells, *L. monocytogenes* uses mainly two members of the surface-exposed leucine-rich repeat (LRR) proteins of the internalin family, namely internalin InlA and InlB, that bind to E-cadherin and hepatocyte growth factor (HGF) Met receptors on the surface of host cells, respectively [59,98,99,100]. Such interaction between the pathogens and the host does not cause significant intestinal inflammation or damage of the intestinal barrier [44]. Internalin A is a cell wall-anchored 80 kDa protein composed of 800 amino acids with a leucine-rich repeat domain with 15 LRRs [90]. Furthermore, the *inlA* gene contains a sequence encoding the signaling peptide at the N-terminal and the LPXTG motif at the C-terminal [43]. *L. monocytogenes* possessing the truncated *inlA* gene, which results in the shorter length of InlA protein, showed a reduced virulence potential compared to the strains with full-length InlA [101]. The intact internalin sequence has been identified in the vast majority of the clinical *L. monocytogenes* strains, whereas the *inlA* truncated gene is more frequent among strains of food origin [102,103,104]. On the other hand, there are also reports which show that *L. monocytogenes* isolates with the truncated *inlA* gene and expressing the non-functional internalin A possess an invasion efficiency in Caco-2 cells and a high pathogenic potential for humans [96,103,105,106].

E-cadherin, the adhesion molecule for InlA, is localized at the basolateral surface of the small intestine cells, including enterocytes localized at villi and junctions between mucus-secreting goblet cells [107]. In enterocytes, E-cadherin is mainly present at adherence junctions and below tight junctions; therefore, it is not exposed to the intestinal lumen [108,109]. E-cadherin is composed of 882 amino acids with 3 main domains: a 555-amino-acid-long N-terminal extracellular part, a transmembrane domain, and a short cytoplasmic part with 152 amino acids [110]. InlA interaction with E-cadherin induces actin cytoskeleton re-arrangements and β- and α-catenins, vezatin, myosin VIIA, Arp2/3, cortactin, and clathrin-mediated endocytosis, resulting in *L. monocytogenes* internalization [95,111].

Attachment of the internalin A-expressing bacterial cells to E-cadherin allows *L. monocytogenes* internalization into enterocytes and, subsequently, translocation across the intestinal barrier. Using cell line and animal models, it has been shown that internalin B is also involved in bacterial internalization at small intestine villi, mainly through promoting endocytosis of junctional components [61]. Although the small intestine is the main site for *L. monocytogenes* host invasion, experiments with transgenic mice demonstrated that bacterial InlA-dependent translocation was also observed at the cecum and the colon of the large intestine [112].

The second important internalin responsible for *L. monocytogenes* binding to host cells is internalin B, encoded by the *inlB* gene located downstream of the *inlA* sequence [60,61,113]. The *inlAB* operon is upregulated during *L. monocytogenes* passages through the intestinal lumen [114]. It binds by interactions to the c-Met cell receptor, with the collaboration of gC1qR and glycosaminoglycans as co-receptors that are involved in stabilization of the junction between InlB and c-Met [115,116]. The c-Met receptor is present on different host cells, which allows *L. monocytogenes* to adhere via InlB to various sites [110]. Thus, InlA exhibits a more restricted cell tropism because the internalin A receptor E-cadherin is mostly present at epithelial cells [110].

Expression of the *inlAB* locus is regulated by the transcriptional regulator PrfA [89,117]. Additionally, activation of *inlA* may be regulated by the general sigma factor σ^B^ regulon via the σ^B^-dependent promoter sequence located upstream of the *inlA* gene [118]. It has been shown that internalin B promotes invasion of intestinal epithelial cells by *L. monocytogenes* when InlA–E–cadherin interactions are functional [112,119]. However, there is also information that InlB plays an important role in this step of bacteria invasion in the absence of functional InlA [61,119,120,121].

The internalin proteins are secreted through the SecYEG translocase that transports proteins across and into the cytoplasmic membrane, and have an amino-terminal cap region followed by an LRR domain [122,123,124]. Additionally, several internalins possess an adjacent inter-repeat (IR) domain that is essential for the binding of the LRR domain to E-cadherin [44,123].

In *L. monocytogenes* adhesion to host epithelial cells, the Peyer’s patches are also involved [96]. As shown in the mouse model, this process is InlB-dependent [60,113]. However, *L. monocytogenes* expressing a murinized InlA, which interacts with N-cadherin expressed in Microfold (M) cells, exhibits increased invasion to this target [96]. *L. monocytogenes* passes through the Peyer’s patches through M cells in an InlA-independent manner and infects CX3CR1^+^ myeloid cells, stimulating the expression of interleukins IL-12 and IL-23 [125]. Subsequently, IL-22-/IL-11-dependent epithelial proliferation is activated, which leads to a decrease in the number of mature goblet cells and locks InlA-dependent pathogen entry through intestinal villi [126]. Following adhesion and internalization into M cells, bacteria are able to spread to the adjacent enterocytes [127].

*L. monocytogenes* may express, other than InlA and InlB, surface-associated and secreted molecules that play a role in adhesion and entry into host cells. Examples of these are the internalins InlE, InlG, and InlH that support the InlA-dependent invasion and modulate the bacterial cell wall organization, and consequently affect InlA exposure [128]. Other putative listerial virulence factors that may modulate cell adhesion and invasion processes are surface adhesins or invasins Ami, Auto, IspC, ActA, Vip, and others, whose roles in pathogenesis of listeriosis have not been deeply clarified [69,129].

Recently, Cahoon et al. [130] described the two-component system PieRS regulates secretion of chaperones PrsA1 and PrsA2 (positive regulatory factor A). It was previously shown that PrsA2 plays a role in invasion, replication, and cell-to-cell spread of *L. monocytogenes* within the infected host [131,132]. In contrast, PrsA1, although showing 75% amino acid similarity with PrsA2, is probably not involved in pathogenesis of *L. monocytogenes* infection [133].

Several other protein molecules also play a role in the *L. monocytogenes* initial step of the host cells’ invasion. Among them, the most important seems to be listeriolysin O cytotoxin, which is involved in bacterial escape to the cytoplasmic space [134]. Furthermore, LLO also has a positive influence on *L. monocytogenes* cell invasion in InlA/InlB-dependent and -independent manners [135]. It has been experimentally shown that listeriolysin O is involved in the induction of *L. monocytogenes* entry to the HepG2 and HeLa cell lines in the absence of InlA or InlB signaling [136].

### 2.3. Epithelial Cell Invasion

*L. monocytogenes* adhered to the E-cadherin or c-Met host receptors is then covered by a phagocytic vacuole in macrophages and enters cells [117]. *L. monocytogenes*, unlike other bacteria, is not destroyed inside the host cell vacuoles due to the production of endogenic factors, mainly listeriolysin O (LLO), a pH- and cholesterol-dependent toxin with pore-forming activity, and GILT (Gamma-interferon Inducible Lysosomal Thiol reductase), found inside the phagosome, which mediate vacuole degradation and bacteria escape to the cytosol [67,137]. LLO is a protein of 56 kDa molecular weight belonging to the cholesterol-dependent cytolysins (CDCs) protein family [138] It effectively binds to lipid membranes characterized with a high concentration of cholesterol [138]. Afterwards, LLO monomers oligomerize to complexes and then undergo a major conformation change that allows them to penetrate the cell membrane and make pores [139]. LLO induces rapid intake of calcium ions inside the host cell and potassium ions efflux, and this process is continuously expressed during the whole intracellular lifecycle of *L. monocytogenes* [140]. Mutants lacking LLO are unable to escape from the phagosome and, consequently, are unable to grow intracellularly [141,142].

There are also mechanisms other than LLO-dependent mechanisms that modulate the *L. monocytogenes* vacuolar infection stage. Bacteria may secrete a peptide pPplA (peptide Pheromone-encoding lipoprotein A) that enhances their escape from host cell vacuoles by activation of the production of an unknown factor that cooperates with LLO in vacuolar damage [68]. Another putative vacuole-related factor is a small GTPase Rab5a, which was shown to control the accelerated maturation of *L. monocytogenes*-containing vacuoles [143,144].

Escaping from the vacuole, a bacterium uses host sugars for its survival and multiplication [3]. Recently, Cheng et al. [140] demonstrated that the listeriolysin O pore-forming activity also involves the phosphorylation of extracellular signal-regulated kinases 1 and 2 in human intestinal epithelial cells infected with *L. monocytogenes*. The whole process begins immediately after the entry of the bacteria into the host cell, and consequently delays maturation of vacuoles, prevents acidification, and allows replication of *L. monocytogenes* [145]. Finally, membrane damage of the phagocytic vacuole allows escape of the bacteria to the cytosol and further spreading to adjacent host cells [145,146].

After release from the vacuole, *Listeria* multiplies and then passes through the host cytosol by polymerizing host actin in an actin assembly-inducing protein (ActA)-dependent manner [69,95,147]. Polymerization of host cell actin during InlA- and InlB-dependent entry is mediated by the Actin-related protein (Arp2/3) complex [148]. This complex is activated by the GTPase-activating protein Rac1, together with the nucleation promoting factors (NPFs) cortactin, or WAVE [148,149]. After that, the actin comet tails are formed, which allow bacteria motility in the cytosol and spread to the adjacent cells by the formation of membrane protrusions that are directly responsible for *L. monocytogenes* transfer into neighboring cells [70,95]. In the latter process, internalin C (InlC), the secreted protein expressed in higher amounts inside infected cells, is involved [95]. It has been shown that the expression of InlC increases during internalization of bacteria into host cells as a result of the *L. monocytogenes* transcription factor PrfA action [71]. InlC interacts directly with the SH3 domain of Tuba, a 177 kDa cytoskeletal protein that can bind various actin regulatory proteins and control the structure of apical junctions [71]. The Tuba protein is also involved in cell junction regulation, cell morphogenesis, and exocytosis [150,151,152,153]. Furthermore, InlC blocks the activation of the NF-kB factor of B lymphocytes, which slows down the response of the host innate immune system [154].

The novel *L. monocytogenes* internalins InlP1 and InlP4, as well as internalin-like protein InlP3, have been identified in strains responsible for a serious listeriosis outbreak in Austria, Germany, and the Czech Republic [64]. Expression of these proteins was increased under gastric stress conditions and in bacteria grown in human intestinal epithelial cells in vitro. Furthermore, InlP1 and InlP4 contributed to the colonization of the spleen and the liver in orally infected mice [64].

Recently, Ling et al. [65] have shown that another *L. monocytogenes* internalin, InlF, contributed to bacteria adhesion and invasion of macrophages, and suppressed the expression of pro-inflammatory cytokines interleukin (IL)-1b and tumor necrosis factor (TNF-a). Moreover, InlF contributed to *L. monocytogenes* colonization in the spleen, liver, and ileum during the early stage of mouse infection, inducing severe inflammatory injury. This suggests that InlF plays a crucial role in modulating the host immune response, contributing to survival in macrophages, and colonization in the early stage of infection [65].

Another way to cross the intestinal epithelium for *L. monocytogenes* involves the interaction of *Listeria* adhesion protein (LAP), a 104 kDa enzyme alcohol acetaldehyde dehydrogenase, with its host cell receptor, the heat shock protein 60 (Hsp60) [62,155]. This LAP receptor is mainly localized at the apical domain of the plasma membrane of ileal villi enterocytes [62]. The LAP protein is present on the bacterial cell wall, but it is also secreted out of the cell by the SecA2 system [156]. It has been shown that the SecA2-deficient mutant showed a reduced cell-to-cell spread in vitro and it was rapidly cleared from the host in the murine model in vivo [157]. The secreted form of LAP, in connection with the cell wall-localized form, promotes full LAP-mediated interaction of *L. monocytogenes* with host epithelial cells and translocation of the bacteria through the epithelium surface [62,63]. The LAP adhesion binding to the heat shock protein 60 (Hsp60) receptor induces secretion of nuclear-kappa B factor (NF-κB), a transcription factor that plays a key role in the production of cytokines IL-6 and TNF-α and the activation of myosin light chain kinase (MLCK) [155,158]. This process promotes cellular redistribution of occludin, claudin-1, and E-cadherin, and, finally, distortion of the tight and adherent cell junctions [158]. These changes decrease the strength of the epithelial layer, allowing translocation of *L. monocytogenes* from the intestinal lumen to the lamina propria [158]. It has been suggested that the LAP-Hsp60 adhesion is the one of the most important steps in crossing the epithelial barrier at the early stage of *L. monocytogenes* infection (12–48 h), whereas the InlA-E-cadherin binding pathway is more relevant for subsequent disease development [43,158].

### 2.4. Dissemination in the Host

After crossing the intestinal barrier and multiplying in the small intestinal lamina propria, *L. monocytogenes* disseminates to the host organs such as the liver, the spleen, and the mesenteric lymph nodes [44] (Figure 2). Furthermore, the bacteria have the ability to cross the blood–brain and fetoplacental barriers [3]. The majority of the cells are trapped in the liver, cleared from the blood circulatory system, and then inactivated through the host immune system, mainly with professional liver phagocytes (Kupffer cells), other mononuclear phagocytic cells, neutrophils, dendritic cells, and natural killer (NK) cells [45]. Furthermore, inactivation of *L. monocytogenes* by the Kupffer macrophages induces monocyte recruitment and a type-1 antimicrobial inflammatory response as well as synthesis of IL-33 by hepatocytes [159]. This process further induces IL-4 expression in basophils and there is a switch from an inflammatory type-1 to a type-2 response, which decreases the host inflammation responses and returns the liver to homeostasis [159]. The remaining live *L. monocytogenes* bacteria are able to replicate in hepatocytes, and then further disseminate within the liver using the actin-mediated cell-to-cell spread pathway [21]. In response to this infection stage, hepatocytes produce chemoattractants that stimulate neutrophils, which results in the induction of growth of typical multifocal granulomas in the liver parenchyma [21]. In addition, liver cells can promote monocyte recruitment via toll-like receptor 2 (TLR2)-dependent secretion of CCL2 and CXCL1 chemokines, which stimulate formation of micro-abscesses and phagocytosis of the bacteria, finally inhibiting their spread [160].

In the spleen, *L. monocytogenes* is mainly engulfed by the macrophages of the marginal zone and then spread to the red pulp [161]. These macrophages interact with CD8α^+^ dendritic cells in the transferring of bacteria into T cell zones and the induction of host immune responses [125].

### 2.5. Cell-to-Cell Spread

After bacteria-containing finger-like protrusion membrane structures of more than ten microns long are formed, *L. monocytogenes* is ready to spread mainly to adjacent cells and, to a much less extent, to other non-adjacent cells [162,163,164]. The bacteria present at the protrusion tips due to the force generated by actin polymerization penetrate the cytoplasm of adjacent cells and spread inside the host [70,95]. It is not clear how *L. monocytogenes* is able to move tens of micrometers to infect non-adjacent host cells, but it is suggested that the bacteria might form protrusions between host cells by disrupting cell–cell junctions [163]. Direct cell-to-cell transfer of *L. monocytogenes* allows the pathogen to resist the humoral and the cytotoxic T cell host immune responses [165]. Then, the recipient cell engulfs the protrusions with the bacteria and allows *L. monocytogenes* to move to the cell cytoplasm [163]. To escape the double-membrane protrusion origin vacuole, the bacterium uses pore-forming toxins and enzymes, mainly listeriolysin O (LLO) and phospholipases A and B (PlcA and PlcB) [166]. LLO is called a phagosome-specific lysin because it has a low and limited activity in the cytosol of host cells and is stable at neutral pH [167,168]. It has been shown that LLO, together with phosphatidylinositol-specific phospholipase C (PI-PLC), are secreted in biologically active extracellular vesicles with diameters ranging from 20 to 200 nm [169]. Activation of listeriolysin O depends both on phagosome pH, with an optimal pH of 5.5, and on GILT, a thiol reductase found inside the phagosome [67,167,170]. The main role of LLO is the destruction of the cell vacuoles to release *L. monocytogenes* into the cytoplasm [66]. However, this listeriosin is also important for downregulation of the host immune system through dephosphorylation of H3 and deacetylation of H4 histones in the cell [171,172].

## 3. Infection Sites

*L. monocytogenes* cells that are still remaining in the circulatory system are rapidly cleared through the host innate immune system, mainly with neutrophils, dendritic cells, and macrophages, although this inactivation process is usually less effective than in the liver [21,173]. Consequently, in immunocompromised persons or persons with a severe immune-deficient system, bacteremia may develop, and *L. monocytogenes* disseminates to a remote organ (e.g., the brain) or crosses the placental barrier in pregnant women [96].

### 3.1. Listeria in the Brain

How the microorganism enters the brain is still not fully clear, mainly due to limited in vivo data available [174]. It has been shown so far that *L. monocytogenes* infiltrates the brain either directly from blood or the nervous cell fibers connected to peripheral tissues [34,175] (Figure 2).

Infection of the central nervous system in humans manifests as meningitis and meningoencephalitis, which are the most frequent clinical presentations observed [176]. Furthermore, brain stem infection (rhombencephalitis) and brain abscessation may develop [34,176]. Meningitis (meningoencephalitis) mainly occurs in persons such as elderly and immunosuppressed patients [37,177,178]. In these cases, it seems that the bacteria inducing this form of infection enter the brain via the hematogenous route [34]. In more detail, *L. monocytogenes* present free in the blood stream crosses the blood–brain barrier, recognizing the specific receptors (e.g., E-cadherin, Met) at the surface of the barrier, and then adheres (to, e.g., InlA, InlB) and passes through it [34]. Alternatively, the bacteria present intracellularly (e.g., inside leucocytes) directly enter the CNF with these infected cells [34,179]. It has been also shown that another *L. monocytogenes* surface-expressed virulence protein factor (Vip) can interact with its Gp96 receptor present at the brain microvessel surface [72]. In a mouse model, the *vip*-deleted *L. monocytogenes* strain was less virulent, and the number of bacteria in the brain was significantly lower compared to the *vip*-positive wild strain [72].

Ghosh et al. [73], using BALB/c mice, demonstrated that one of the internalin family surface proteins, InlF, plays a role in *L. monocytogenes* colonization of the brain through binding to the host cell receptor vimentin. Since vimentin is broadly expressed on the surface of brain microvascular endothelial cells and astrocytes, it may be important in the adhesion step of *L. monocytogenes* during brain invasion [180,181,182].

Another way of *L. monocytogenes* brain entry is the mentioned above transportation of the bacteria inside infected phagocytic cells such as monocytes and dendritic cells or bone marrow myelomonocytic cells directly to the CNS (so-called Trojan horse mechanism) [183,184]. This process has been shown as interferon-γ-dependent, but chemokine receptor type 2 (CCR2)-independent [185]. Afterwards, the bacterium is able to infect brain neurons spreading from these *L. monocytogenes*-carrying cells [186].

Recently, Maudet et al. [74] developed a clinically relevant humanized mouse model of neurolisteriosis using hypervirulent *L. monocytogenes* strains and showed that monocytes are necessary and sufficient to induce neuroinvasion. It has been also documented that InlB possesses a major role in this process, whereas InlA is not involved in neuroinvasion [74]. InlB protects infected monocytes from Fas-mediated cell death by CD8^+^ T cells in the c-Met, PI3 kinase, and FLIP-depending manners, finally increasing their lifespan and adhesion to brain vessels, and thereby favoring the transfer of *L. monocytogenes* from infected monocytes to the brain [74].

In the process of brain stem infection (rhombencephalitis), which in humans is identified in up to 24% of patients, *L. monocytogenes* enters the organ mainly via neural retrograde transport [37,187]. It has been suggested that the bacteria may enter the host during mucosal injury of the oropharyngeal and nasal cavities, lips, conjunctiva, or gut, i.e., as a result of surgical procedures [188,189]. It seems that the trigeminal nerve pathway represents the most efficient way by which *L. monocytogenes* invades the brain stem from the oropharynx [190]. However, there are rather few experimental reproductions of typical rhombencephalitis lesions in animals such as sheep and goats [7,191,192].

Once in the brainstem, *L. monocytogenes* spreads further to brain center and caudally to the spinal cord along axonal connections of trigeminal nerve branches and other cranial nerves [187]. The way in which trigeminal and other peripheral nerves become infected prior to further dissemination to the CNS remains unknown [34]. It has been suggested, based on the studies with ruminants, that in this process an InlA-dependent mechanism of invasion of cranial nerves plays a role [44]. In this animal model, *L. monocytogenes* adheres to E-cadherin expressing oral epithelial cells or myelinating Schwann cells, and subsequently spreads to neighboring axons [193]. Furthermore, the source of axonal infection may also be phagocytes by ActA-dependent bacterial cell-to-cell spread [186]. Some experiments in mice have also shown the role of phospholipase B (PlcB) in the dissemination of *L. monocytogenes* bacteria from peripheral macrophages to the trigeminal nerve [190].

Brain infection with *L. monocytogenes* causes meningitis or meningoencephalitis, with headache, fever, and neck stiffness [28,174,194]. These signs of the disease cover the majority of CNS infections by *L. monocytogenes* among people (70–97%) [37,177]. There are predisposing factors such as immunosuppression, age over 50 years, malignancy, or diabetes [177]. In some patients (below 10%, mostly immunocompromised), macroscopic brain abscesses located in subcortical areas, thalamus, pons, or medulla have been observed [195].

Further development of the CNS infections results in rhombencephalitis, which occurs in up to 24% of listeriosis patients and is characterized by progressive brain stem dysfunction [34,37,176,196]. During the first days, unspecific symptoms are developed such as headache, malaise, nausea, and vomiting. After that, asymmetrical cranial nerve deficits, cerebellar signs, hemiparesis, and hemisensory defects may develop, frequently together with meningeal symptoms [34]. Interestingly, rhombencephalitis is mainly seen in healthy persons without any other diseases or in patients with an impaired immune system [34,196,197,198].

### 3.2. Listeria in Fetus

*L. monocytogenes* is also able to cross the placental barrier and infect the fetus in pregnant women, resulting in still-birth or frequent lethal neonatal infections [20,199,200] (Figure 2). The whole entry process is not fully clarified, but it seems that bacteria may cross the endothelium of the maternal blood vessels and then pass into the fetal circulatory system of the placental villi [21,96]. Infection of the fetus may be achieved in two ways: either as a cell-to-cell spread from maternal infected phagocytes, or via infection of trophoblasts with *L. monocytogenes* that are circulating in the blood [96]. One of the most important elements in the barrier between the mother’s blood and her fetus are cytotrophoblasts, the inner layer of villous trophoblasts, which are considered to be stem cells for the syncytiotrophoblast, which is in direct contact with maternal blood in the villous human placenta [201,202]. Furthermore, the extravillous cytotrophoblasts anchor the villous tree in the decidua [201]. Using immunohistochemical analysis, it has been shown in vitro that *L. monocytogenes* is able to infect placentas obtained from women with listeriosis [203]. Furthermore, most infections are noted late in the pregnancy, suggesting that the syncytiotrophoblast may be the main entry point of the bacteria in the placenta [178]. Broad studies have shown that *L. monocytogenes*, after passing beyond the barrier of invasive extravillous trophoblasts that inhibits the vacuolar escape of the bacterium, colonizes the decidua of the endometrium [204]. Several investigations also demonstrated that infection of the placenta with *L. monocytogenes* is dependent on the number and virulence potential of circulating bacteria that will resist the protective barrier of extravillous cytotrophoblasts and syncytiotrophoblasts [205,206,207,208].

The cellular mechanism of syncytiotrophoblast infection with *L. monocytogenes* is different from the invasion of enterocytes because interactions of InlA and InlB with their respective receptors, E-cadherin and c-Met, are necessary for efficient gut tissue invasion [209]. However, unlike enterocytes, syncytiotrophoblasts have no intrinsic phosphoinositide 3-kinase (PI3-K) activity required for bacteria internalization [112,210]. Binding of internalin B to c-Met induces PI3-K activity that, in turn, phosphorylates the plasma membrane lipid second messenger phosphoinositide-4,5-bisphosphate (PIP2) into phosphoinositide-3,4,5-trisphosphate (PIP3), which is essential for *L. monocytogenes* internalization [211]. On the other hand, using the gerbil animal model, it was demonstrated that bacterial mutants lacking the ability to express InlA and InlB can still infect the fetus, which suggests that in the placental invasion process, at least partially, maternal circulating phagocytes are involved [112].

Faralla et al. identified the *inlP (lmo2470)* gene encoding one of the internalin family proteins, a secreted internalin P (InlP) [75]. This internalin is conserved in virulent *L. monocytogenes,* but is absent in strains that are nonpathogenic for humans. Furthermore, it has a strong tropism for the placenta and studies with guinea pig and mice models demonstrated that deletion of the *inlP* gene provided a 1000-fold defect in placental colonization [75]. It has been shown that InlP binds to the host cell cytoplasmic protein afadin, a protein associated with cell–cell junctions [212]. The InlP–afadin interaction specifically enhances *L. monocytogenes* transcytosis through the basal face of polarized epithelial cells, resulting in placenta infection [213].

Pregnant women infected with *L. monocytogenes* may be asymptomatic or show unspecific flu-like symptoms, but clinical manifestations such as meningoencephalitis or endocarditis have not been often reported [29,214]. However, the pregnancy-associated infection usually presents in the fetus with septicemia, pneumonia, or meningitis, and often finalizing with abortion (20–30% cases), giving birth to a still child, or premature birth [214,215]. Some studies demonstrated that not all listeriosis cases of the mothers during pregnancy are associated with infection of the fetus, and ca. 30% of pregnancy-related cases do not result in any of the described above complications [37,216,217,218].

The neonatal form of listeriosis is usually the result of vertical transmission of *L. monocytogenes* from mother to fetus, either by ingestion of infected amniotic fluid during intrauterine life, transplacentally from the maternal circulation or by ascending colonization from the infected vagina, or, rarely, by horizontal infection after birth [214,219]. In the USA, neonatal listeriosis has been identified in approximately 8.6/100,000 live births, with a high (20–60%) mortality rate [214,220].

Neonatal listeriosis may manifest in two forms, depending on the time of development of symptoms after birth, e.g., as early and late onset [219,221]. The early form develops at days 1–6 of life, and it is usually associated with previous mild maternal symptoms [219]. In newborns, in most cases septicemia develops (80–81% of children), and in some of them respiratory disorders or pneumonia, and sometimes meningitis, are identified [222]. Additionally, abscesses and granulomas arise and disseminate in multiple organs (e.g., liver) as well as severe neurological complications in the surviving neonates may develop [219,223]. The mortality rate of early neonate listeriosis is estimated at ca. 20% [219].

The second form of neonatal listeriosis (late onset *L. monocytogenes* infection) develops in infants at ages 5 to 20 days who were born to asymptomatic mothers with usually an uneventful, carried-to-term pregnancy [214]. Such newborns have no disease symptoms at birth, but after a few days meningitis develops, sometimes connected with fever, colitis, and diarrhea [45,214,217,224]. The neonates are usually infected through contact with the *L. monocytogenes*-contaminated birth canal, maternal feces, or the home environment [45,224]. The fatality rate of this form of neonatal listeriosis is lower than during the early onset disease, and is estimated at ca. 10%. However, often various severe complications in surviving infants such as growth retardation, intellectual disability, and blindness are observed [32].

## 4. Treatment of Listeriosis

*L. monocytogenes* is susceptible to many different antimicrobials that are used for gram-positive bacteria [225,226]. These include β-lactams, gentamicin, erythromycin, tetracycline, rifampicin, and vancomycin. On the other hand, *L. monocytogenes* is naturally resistant to cephalosporins, nalidixic acid, and polymyxin E [225]. Furthermore, most isolates are not susceptible to fluoroquinolones and cephalosporins of the third (e.g., cefotaxime) and fourth (cefepime) generations, nor to fosfomycin, oxacillin, and lincosamides [227]. High resistance to tetracyclines has been also reported in a few strains [225,228]. Antimicrobial resistance/susceptibility of *L. monocytogenes* strains varies widely and depends on sampling sites, time of sampling, source of isolates, and geographical origin.

In 1988, the first multiresistant (i.e., showing resistance to three or more classes of antimicrobials) *L. monocytogenes* strain of human origin was described in France [229]. It displayed resistance to chloramphenicol, erythromycin, streptomycin, and tetracycline. The genes responsible for resistance to these antibiotics were located on a 37 kb plasmid. Since then, several other *L. monocytogenes* strains with antimicrobial multiresistance patterns have been isolated from different clinical, food, and environmental origins [228,230,231,232,233].

Based on *L. monocytogenes* in vitro antimicrobial resistance results, treatment of severe listeriosis with β-lactams (penicillin or ampicillin) alone or combined with an aminoglycoside (e.g., kanamycin or gentamicin) is recommended [230,234]. Additionally, in cases of reduced sensitivity or resistance of the strains to β-lactams, other antimicrobial substances that are effective against gram-positive bacteria may be used, e.g., tetracyclines, erythromycin, chloramphenicol, vancomycin, and trimethoprim/sulfamethoxazole [234]. The latter antimicrobials are recommended for patients with an allergy to penicillin, whereas persons with bacteremia due to *L. monocytogenes* may be treated with vancomycin [234]. Furthermore, erythromycin can be used in patients with an allergy to ampicillin and/or gentamicin [214]. During pregnancy of *L. monocytogenes*-infected women, ampicillin or erythromycin intravenously or amoxicillin orally are used for at least 14 days, or even until delivery [224]. When the woman does not tolerate penicillin or amoxicillin, trimethoprim with sulfamethoxazole are the drugs of choice [224]. However, trimethoprim can damage the fetus in the early stages of pregnancy, including its heart and nervous system; therefore, patients with penicillin intolerance expecting a baby can be treated with erythromycin that is safe for fetus [235,236].

An increasing resistance of *L. monocytogenes* against antimicrobials, including antibiotics, led to the search for alternative therapies [237]. One of them is the application of bacteriocins, which are natural peptides produced by various bacteria [238,239]. Several bacteriocins are stable in gastrointestinal conditions, possess a low toxicity, and show a significant effect against pathogenic bacteria, including antibiotic-resistant strains [240,241]. It was shown that nisin (produced by *Lactococcus lactis*) and pediocin (secreted by *Pediococcus acidilactici***)**, demonstrated inhibitory activity against *L. monocytogenes* both in vitro and in vivo [242,243].

Another group of antibacterial substances that are promising alternatives to antibiotics are natural products of plant origin that possess different mechanisms of action directed towards increasing membrane permeability, decreasing its integrity, or disruption of bacterial efflux pumps [244]. Among them are terpenoids such as limonene and carvacrol, which has been shown to be effective against *L. monocytogenes* [245,246].

The efficacy of other phytochemicals such as trans-cinnamaldehyde, carvacrol, and thymol in reducing *L. monocytogenes* virulence was demonstrated using the *Galleria mellonella* invertebrate model [247].

Although all these plant products demonstrated antibacterial activity, their application in treatment of listeriosis requires further mammalian and clinical studies [244].

## 5. Prevention of Listeriosis

Currently, there is no effective vaccine against listeriosis, although some experiments have shown that cell-based and subunit-based immunoprepartions lacking cytotoxicity and pathogenicity may be highly protective [248,249,250]. Furthermore, *L. monocytogenes* is often applied as a vaccine vector for protection against other pathogens as well as in cancer therapy [251,252]. Therefore, prevention of *L. monocytogenes* infection is the most important way to control the disease, since the bacterium is widely distributed in the environment, including in food-production facilities [3,253]. It has been revealed that *L. monocytogenes* is able to survive there for a long time due to inadequate cleaning and disinfection of food production equipment or insufficient supervision of employees [17]. Most sporadic listeriosis cases and large outbreaks were due to the consumption of contaminated food with these bacteria, especially ready-to-eat food of animal origin [17]. Listeriosis is a typical zoonotic food-borne disease, although it may be also transmitted through direct contact with infected animals or contaminated environments [21,22].

There are studies on the prevention of *L. monocytogenes* infection by the use of probiotics [254,255,256,257,258]. Probiotic bacteria, e.g., lactobacilli, have shown positive effects in mice by producing bacteriocin or by changing the host gene expression or *L. monocytogenes* transcriptome [259]. Drolia et al. demonstrated that *Lactobacillus casei* expressing *Listeria* adhesion protein (LAP) colonized the intestine, adhered to the heat shock protein 60 (Hsp60) receptor, and excluded *L. monocytogenes* from intestinal colonization and systemic dissemination [257]. Furthermore, such probiotic bacteria were also able to prevent fetoplacental transmission of the pathogen in a pregnant guinea pig model [258]. Another study of Mathipa et al. showed that in *L. casei* expressing *L. monocytogenes,* InlA and InlB inhibited adhesion, invasion, and translocation of *L. monocytogenes* through enterocyte-like Caco-2 cells [256]. All these results suggest that molecularly recombinant probiotic bacteria possessing *Listeria* virulence traits may be a potential approach for prevention of human listeriosis.

The key aspect in prevention of listeriosis is proper food preparation, handling, and storage to avoid its contamination or cross-contamination and then consumption by humans [260,261,262]. According to the European Union food law regulation, the criteria for *L. monocytogenes* varies on the food category and intended consumer population [263]. Generally, there is zero tolerance for ready-to-eat foods for infants and for food of special medical purposes, and up to 100 cfu/g for other ready-to-eat foods.

Crucial factors for listeriosis prevention cover rapid and specific detection of *L. monocytogenes* in food with classical or alternative methods and determination of the infection sources [264]. *L. monocytogenes* is not only able to persist, but also multiply in a wide range of adverse conditions present in food production environments and create biofilms on various surfaces. All these features make it difficult to eliminate the bacteria and enable them to survive there for a long time [265]. Therefore, efficient methods of removal of the pathogen from the food industry environment, which are fundamental for ensuring the safety of food production, should be developed and applied [266].

## 6. Conclusions

Listeriosis, although it is a rather uncommon disease compared to many other food-borne illnesses, is one of the most severe infections due to its high frequency of deaths of approximately 20–30%. The disease is caused by *L. monocytogenes*, the ubiquitous bacterium present in soil, environments, and in food products, that has the ability to survive and grow under unfavorable conditions often found in food production plants. Furthermore, it expresses several factors ensuring survival in unfavorable gastrointestinal conditions. It is also an unusual pathogen because it is the intracellular microorganism that causes various types of the disease characterized as septicemia and encephalitis. *L. monocytogenes* has also placental tropism resulting in the infection of the fetal tissues and finally the death of the child. The bacterium possesses several virulence factors, responsible for epithelial cell adhesion, cell-to-cell spread, intracellular multiplication, and crossing of natural host barriers. Among them are internalins, including the recently described InlF. It is well known that invasion of the host non-phagocytic cells is critical in the pathogenesis of listeriosis. Various bacterial surface molecules involved in the invasion of host cells have been identified and showed that *L. monocytogenes* has developed complex strategies of affecting many different tissues and organs. However, there are also gaps in the understanding of specific interactions between host factors and the microorganism on its pathogenicity. Thus, further studies towards *L. monocytogenes* investigation of molecular pathogenic potential and mechanisms of the infection are needed.

## Figures and Tables

**Figure 1 pathogens-11-01491-f001:**
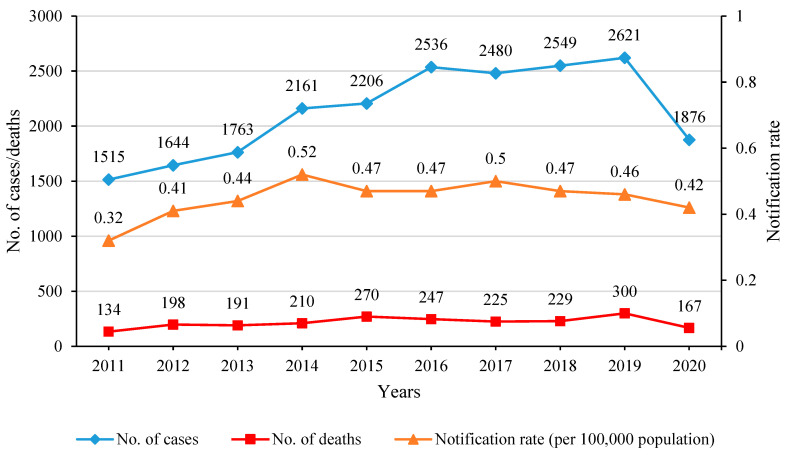
Incidence of listeriosis in humans in the European Union countries between 2011 and 2020. In 2020 data from the United Kingdom, as non-EU members were not provided.

**Figure 2 pathogens-11-01491-f002:**
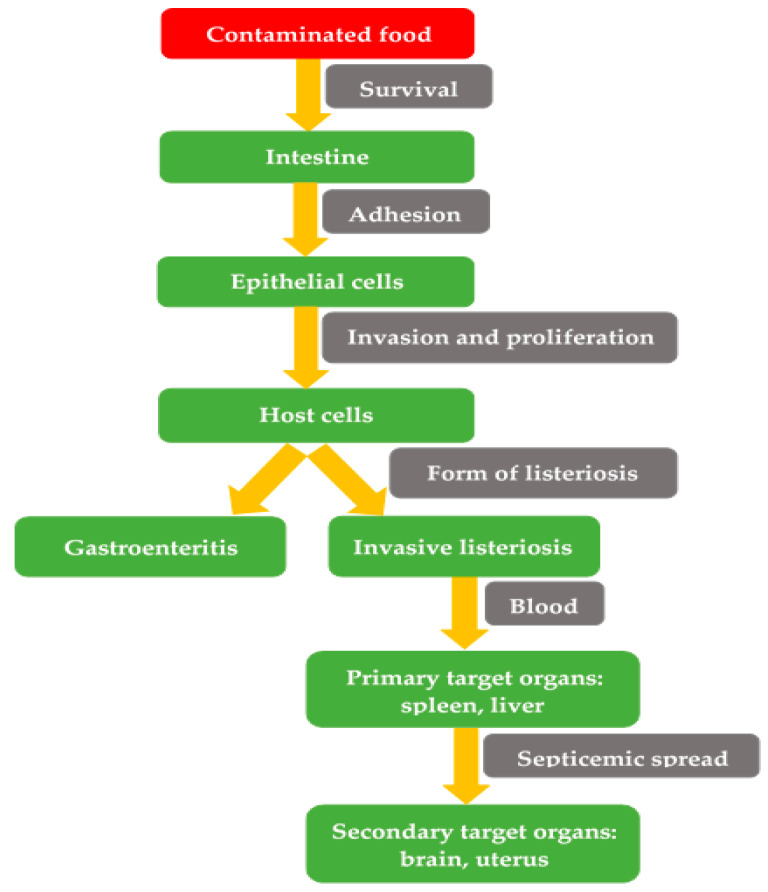
Schematic *L. monocytogenes* infectious cycle in the human host. Bacteria enter the host through contaminated food and invade the epithelial cells, potentially causing gastroenteritis. Crossing the intestinal barrier, the bacteria spread via blood to their primary target organs (liver and spleen). Then, the bacteria may spread to secondary target organs (uterus, brain), resulting in abortion in pregnant women or meningoencephalitis, respectively.

## Data Availability

Not applicable.

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
