# Peer review of "Listeria monocytogenes—How This Pathogen Uses Its Virulence Mechanisms to Infect the Hosts"

_pathogens, 2022, doi:10.3390/pathogens11121491_

Round 1

Reviewer 1 Report

The manuscript has an interesting and current theme. It is not mentioned in the work what its aim is, although it is a review.

In addition, despite being a review, the inclusion and exclusion criteria, descriptor terms or possible filters used in the preparation of the manuscript were not addressed.

Most references are not recent (last 5 years).

Therefore, it is difficult to judge whether the design proposed in carrying out the review was adequate, as there is no presentation of a methodology that directs us to confirm or not the proposed hypothesis.

Only one figure and one table were presented, which are adequate. But this feature could have been further explored to contextualize the information presented.

The conclusion is apparently a summary of some topics covered in the manuscript.

The review is clear, comprehensive and relevant, but alternative or adjunctive treatments to the drug can be discussed, as was done in the prevention topic.

There are many recent publications on L. monocytogenes and listeriosis. This manuscript may be complementary to some of them, if more recent references are used.

In this journal, 12 reviews were published with the theme of listeria.

In this sense, I suggest that more current and relevant information be added so that the manuscript can be accepted.

Author Response

Dear Reviewer, We thank you very much for your much effort and excellent review process of our manuscript. Your very valuable comments have allowed us to significantly improve the quality of our paper. Below, please find our responses to your remarks.

The manuscript has an interesting and current theme. It is not mentioned in the work what its aim is, although it is a review.

Response: The aim of this review manuscript has been added both in revised Abstract (current lines 19-22) and in Introduction (current lines 27-36).

In addition, despite being a review, the inclusion and exclusion criteria, descriptor terms or possible filters used in the preparation of the manuscript were not addressed.

Response: The present review is based on the most important papers related to L. monocytogenes pathogenesis published until October 2022, which are available on the Web of Science Core Collection. This has been clarified in the revised version of the manuscript (current lines 27-28).

Most references are not recent (last 5 years).

Response: Indeed, several references are older than 5 years but we think that all of them are important for this comprehensive review. However, a total of 58 references are from last 5 years, including 13 new items that were added during revision of the current version of the manuscript.

Therefore, it is difficult to judge whether the design proposed in carrying out the review was adequate, as there is no presentation of a methodology that directs us to confirm or not the proposed hypothesis.

Response: It has been clarified in the revised manuscript how this review was prepared and what literature was selected (current lines 27-28).

Only one figure and one table were presented, which are adequate. But this feature could have been further explored to contextualize the information presented.

Response: Both figure and table have been cited and explained in the text of the manuscript. Furthermore, additional figure (Figure 2) has been added to summarize the pathogenesis of listeriosis.

The conclusion is apparently a summary of some topics covered in the manuscript.

Response: The conclusion has been slightly revised and new sentences have been added (lines 704, 708, 711).

The review is clear, comprehensive and relevant, but alternative or adjunctive treatments to the drug can be discussed, as was done in the prevention topic.

Response: Information related to alternative strategies for treatment of listeriosis have been added (lines 538-555) and the relevant literature have been cited.

There are many recent publications on L. monocytogenes and listeriosis. This manuscript may be complementary to some of them, if more recent references are used.

Response: Thirteen recent publications have been added in the revised version of the manuscript.

In this journal, 12 reviews were published with the theme of listeria.

Response: We have revised the reference list of the manuscript and added three new reviews published in the Pathogens journal that are related to the current manuscript (Lamond and Freitag, 2018; Kawacka et al., 2021; Wei et al., 2020).

In this sense, I suggest that more current and relevant information be added so that the manuscript can be accepted.

Response: We hope that the revised version of the manuscript, that includes all your comments and suggestions, is now acceptable for publication in Pathogens.

Reviewer 2 Report

Minor Comments

Point 1: In this research article, the authors present Listeria monocytogenes Virulence Mechanisms to Infect the Hosts; it would be better to specify some major mechanisms in the abstract.

Point 2: Some of the references are too old. I recommend you revise it where possible. In addition, there are some grammatical errors (from abstract to conclusion), and a strict English reviewer is needed.

Line: 724-725  “Molecularly recombinated” should be molecularly recombinant.

Point 3: Furthermore, the authors need to add one additional figure to describe the fundamental mechanism used by L. monocytogenes during infection, as discussed in this review. Otherwise, it would be difficult for the reader to capture the overall picture of the study.

Author Response

Dear Reviewer, We thank you very much for your much effort and excellent review process of our manuscript. Your very valuable comments have allowed us to significantly improve the quality of our paper. Below, please find our responses to your remarks.

Comments and Suggestions for Authors

Minor Comments

Point 1: In this research article, the authors present Listeria monocytogenes Virulence Mechanisms to Infect the Hosts; it would be better to specify some major mechanisms in the abstract.

Response: The whole Abstract has been revised and some major virulence factors involved in pathogenesis od L. monocytogenes infection have been added. However, the Abstract limit for the Pathogens journal is 200 words and we had to comply with these requirements.

Point 2: Some of the references are too old. I recommend you revise it where possible. In addition, there are some grammatical errors (from abstract to conclusion), and a strict English reviewer is needed.

Response: The present review is based on the most important papers related to L. monocytogenes pathogenesis published until October 2022, which are available in the Web of Science Core Collection. This has been clarified in the revised version of the manuscript (current lines 27-28). Indeed, several references are seem old but they are basic and important for this comprehensive review; thus, we think they should be left in the revised manuscript. However, 13 new published in last years were added during revision of the current version of the manuscript.

Furthermore, the revised manuscript has been read again by the English-native person to avoid any grammatical errors.

Line: 724-725  “Molecularly recombinated” should be molecularly recombinant.

Response: Done.

Point 3: Furthermore, the authors need to add one additional figure to describe the fundamental mechanism used by L. monocytogenes during infection, as discussed in this review. Otherwise, it would be difficult for the reader to capture the overall picture of the study.

Response: Additional figure (Figure 2) which shows schematic L. monocytogenes infectious cycle in the human host has been added.

We hope that the revised version of the manuscript, that includes all your comments and suggestions, is now acceptable for publication in Pathogens.

Reviewer 3 Report

Review article "Listeria monocytogenes – How This Pathogen Uses Its Virulence Mechanisms to Infect the Hosts? is too long and with constant repetitions the emphasis is not on virulence factors. I think that the manuscript is too long and should be reorganized as it is in the title or the title should be changed.

Facts are repeated in epidemiology and pathogenesis, and I don't know if it is necessary to go into so much detail when it comes to pathogenesis and separate, detailed descriptions of individual organs.

There is also a lack of a schematic representation that would connect all that story about virulence factors and the host's response.

The manuscript needs to be completely reorganized.

Author Response

Dear Reviewer, We thank you very much for your much effort and excellent review process of our manuscript. Your very valuable comments have allowed us to significantly improve the quality of our paper. Below, please find our responses to your remarks.

Comments and Suggestions for Authors

Review article "Listeria monocytogenes – How This Pathogen Uses Its Virulence Mechanisms to Infect the Hosts? is too long and with constant repetitions the emphasis is not on virulence factors. I think that the manuscript is too long and should be reorganized as it is in the title or the title should be changed.

Response: The whole manuscript has been deeply reorganized, especially to avoid repetitions.

Facts are repeated in epidemiology and pathogenesis, and I don't know if it is necessary to go into so much detail when it comes to pathogenesis and separate, detailed descriptions of individual organs.

Response: The aim of our review manuscript was to present a comprehensive information on L. monocytogenes, i.e. epidemiology, pathogenesis, treatment and prevention of listeriosis. The review is based on the most important papers related to L. monocytogenes pathogenesis published until October 2022, which are available in the Web of Science Core Collection. This has been clarified in the revised version of the manuscript (current lines 27-28). Thus, the manuscript seems to be long but we hope that all the data included will be suitable for a broad range of readers to better understand of this important microorganisms and different forms of the disease that are caused by L. monocytogenes.

There is also a lack of a schematic representation that would connect all that story about virulence factors and the host's response.

Response: Additional figure (Figure 2) which shows schematic L. monocytogenes infectious cycle in the human host has been added. We hope this picture will connect and clarified different steps of infection.

The manuscript needs to be completely reorganized.

Response: As you suggested, the whole manuscript has been revised and reorganized. We hope that this new version of the manuscript, that includes all your comments and suggestions, is now acceptable for publication in Pathogens.

Round 2

Reviewer 1 Report

After requested adjustments, I believe that the manuscript is ready to be published.

Reviewer 3 Report

The manuscript has been edited according to the suggestions and I have no further comments.